# Molecular Regulatory Network of Anthocyanin Accumulation in Black Radish Skin as Revealed by Transcriptome and Metabonome Analysis

**DOI:** 10.3390/ijms241713663

**Published:** 2023-09-04

**Authors:** Jing Zhang, Zi-Xuan Zhang, Bo-Yue Wen, Ya-Jie Jiang, Xia He, Rui Bai, Xin-Ling Zhang, Wen-Chen Chai, Xiao-Yong Xu, Jin Xu, Lei-Ping Hou, Mei-Lan Li

**Affiliations:** 1College of Horticulture, Shanxi Agricultural University, Taigu, Jinzhong 030801, China; 2Institute of Vegetable, Zhengzhou 450005, China; 3Key Laboratory of Innovation and Utilization for Vegetable and Flower Germplasm Resources in Shanxi, Taiyuan 030000, China

**Keywords:** radish, black root skin, anthocyanins, metabolome, transcriptome, WGCNA

## Abstract

To understand the coloring mechanism in black radish, the integrated metabolome and transcriptome analyses of root skin from a black recombinant inbred line (RIL 1901) and a white RIL (RIL 1911) were carried out. A total of 172 flavonoids were detected, and the analysis results revealed that there were 12 flavonoid metabolites in radish root skin, including flavonols, flavones, and anthocyanins. The relative concentrations of most flavonoids in RIL 1901 were higher than those in RIL 1911. Meanwhile, the radish root skin also contained 16 types of anthocyanins, 12 of which were cyanidin and its derivatives, and the concentration of cyanidin 3-o-glucoside was very high at different development stages of black radish. Therefore, the accumulation of cyanidin and its derivatives resulted in the black root skin of radish. In addition, a module positively related to anthocyanin accumulation and candidate genes that regulate anthocyanin synthesis was identified by the weighted gene co-expression network analysis (WGCNA). Among them, structural genes (*RsCHS*, *RsCHI*, *RsDFR*, and *RsUGT75C1*) and transcription factors (TFs) (*RsTT8*, *RsWRKY44L*, *RsMYB114*, and *RsMYB308L*) may be crucial for the anthocyanin synthesis in the root skin of black radish. The anthocyanin biosynthesis pathway in the root skin of black radish was constructed based on the expression of genes related to flavonoid and anthocyanin biosynthesis pathways (Ko00941 and Ko00942) and the relative expressions of metabolites. In conclusion, this study not only casts new light on the synthesis and accumulation of anthocyanins in the root skin of black radish but also provides a molecular basis for accelerating the cultivation of new black radish varieties.

## 1. Introduction

Anthocyanins are important secondary metabolites of plants [1] and the final product of flavonoid metabolism. As the second-largest pigment in plants, they are widely present in the vacuoles of leaves, flowers, fruits, and other organs. In addition to being responsible for the orange–red to blue–purple coloring of plant organs [2], anthocyanins attract pollinators for the pollination of plants and play an important role in resisting biotic or abiotic stress [3]. More importantly, anthocyanins can promote human health [4] and have become one of the important symbols of plant nutrients or food health.

The radish (*Raphanus sativus* L.) is one of the most important root vegetables belonging to the family Cruciferae [5]. Because the radish originated in China, China enjoys the most abundant radish germplasm resources. After long-term evolution and selection, radish varieties with different root skin colors have been formed, such as those with white, red, purple, yellow, green, or black root skin. Different colors of the root skin are attributed to anthocyanin accumulation [6]. The primary anthocyanidins in the red radish varieties are pelargonidins [7,8,9], while those in the purple or black radish varieties are cyanidins [10,11]. The black radish is rich in anthocyanins, and its root skin is black, or purple–black, which has a high nutritional and economic value [11,12].

Anthocyanins accumulate in different organs of the radish, some in the roots’ flesh, some in the roots’ skin, stem, leaf, or stem and leaf. Previous RNA-seq analysis showed that the mechanism of anthocyanin accumulation differs in different radish varieties [5,6,13,14,15]. In addition, the role of a few genes in anthocyanin synthesis in the red radish has been clarified. *RsMYB1* positively regulates the anthocyanin biosynthesis pathway in the red radish [16]. Heterologous co-expression of both *RsTT8* and *RsMYB1* in tobacco leaves dramatically increases the expression of endogenous anthocyanin biosynthesis-related genes [17]. When *RsMYB1a* is overexpressed alone or together with *RsbHLH4*, large quantities of anthocyanins accumulate in radish cotyledons and leaves [18]. *RsGST1* from the red radish can restore anthocyanin accumulation in *Arabidopsis tt19* mutants, indicating that *RsGST1* has a similar function as *AtTT19*, a gene responsible for the transport of anthocyanins in *Arabidopsis thaliana*. In addition, *RsGST1* and *RsMYB1a* play a synergistic role in anthocyanin accumulation in radish [19].

At present, the germplasm resources of the black radish mainly include deep red, deep purple, purple–black, or black root skin and white, pink, light purple, or purple–black root flesh [20,21]. The difference in anthocyanin accumulation in different parts of the black radish root may be related to the synthesis mechanism of anthocyanins in different materials. However, the molecular regulation mechanism of anthocyanin biosynthesis in black radish is still unclear. Therefore, in this study, a black radish recombinant inbred line (RIL 1901) and a white radish RIL (RIL 1911) were used to detect and quantify flavonoid metabolites in radish root skin by ultra-high performance liquid chromatography-tandem mass spectrometry (UPLC-MS/MS). Meanwhile, the genes regulating anthocyanin biosynthesis were analyzed by RNA-seq, and these gene expressions were further detected by real-time quantitative polymerase chain reaction (qRT-PCR). In addition, co-expression gene modules were obtained by weighted gene co-expression network analysis (WGCNA), and hub genes related to anthocyanin biosynthesis were filtered. Furthermore, the biosynthesis pathway of anthocyanins in the root skin of black radish was constructed, which laid a foundation for further study of the regulation pathway of anthocyanin biosynthesis in the root skin of black radish. In closing, this study clarified the molecular mechanism of anthocyanin accumulation in black radish, enriched the basic theory of plant anthocyanin biosynthesis, and provided a basis for genetic improvement and germplasm innovation of radish.

## 2. Results

### 2.1. Identification of Flavonoids in the Root Skin of Two Radish RILs

To compare the concentration of flavonoids in the root skin of two radish RILs (RIL 1901 and RIL 1911) at different development stages, the root skin samples of RIL 1901 and RIL 1911 were analyzed by LC-MS/MS. A total of 172 types of flavonoids were detected, including 63 flavonols, 39 flavonoids, 16 anthocyanins, 12 flavonoid carbonosides, 12 flavanols, eight dihydroflavones, six isoflavones, five dihydroflavonols, five proanthocyanidins, four chalcones, one dihydroisoflavone, and one other flavonoid (Appendix A). With |Log2FC| ≥ 1, *p* < 0.05, and VIP ≥ 1 as thresholds for differentially accumulated flavonoids (DAFs), the number of DAFs in the two radish RILs ranged from 42 to 55 at different development stages, and there were 60 DAFs at the maturity stage (Figure 1B). These DAFs were mainly distributed in anthocyanins, flavones, and flavonols, and the relative concentration of most DAFs in RIL 1901 was higher than that in RIL 1911 (Appendix A).

Anthocyanins are the most important flavonoid colorants in plants. A total of 16 anthocyanins were detected in the radish root skin, including 12 types of cyanidin, one type of pelargonidin, and three types of peonidin. The black root skin of radish was mainly caused by cyanidin and its derivatives, among which the concentration of cyanidin 3-O-glucoside was the highest in each period of black radish. In the four periods of RIL 1901, the accumulation pattern of cyanidin derivatives was relatively complex, and the concentrations of cyanidin 3-O-glucoside and peonidin 3-O-glucoside in b1 were significantly higher than other anthocyanins, so the two may result in the purple appearance at b1 (Table 1). Cyanidin-3-O-sophorotrioside-5-O-malonylglucoside, cyanidin-3-O-p-coumaroyl-sophoroside-5-O-malonyl-glucoside, cyanidin-3-O-feruloyl-sophoroside-5-O-malonylglucoside, cyanidin-3-O-p-coumaroyl-feruloyl-sophoroside-5-O-malonyl-glucoside, cyanidin-di-O-malonyl-glucoside-O-malonyl-diglucoside, cyanidin-3-O-p-coumaroyl-caffeoyl-sophoroside-5-O-succinoyl-glucoside, and cyanidin-3-O-sinapoyl-5-O-dimalonylglucoside-glucosid showed an accumulation trend at the four stages of the RIL 1901. At b4, pelargonidin 3-O-(6″-acetylglucoside) rapidly accumulated, while the concentration of cyanidin 3-O-(3″,6″-dimalonylglucoside) sharply decreased to almost zero. In addition, the concentrations of cyanidin-3-O-di-caffeoyl-diglucoside-5-O-malonyl-glucoside, cyanidin-3-O-p-coumaroyl-feruloyl-sophoroside-5-O-malonyl-glucoside, cyanidin-3-O-sinapoyl-5-O-dimalonylglucoside-glucosid, cyanidin 3-O-glucoside, cyanidin 3,5-O-diglucoside, and peonidin 3-O-glucoside were low at w4, suggesting that small quantities of anthocyanins are still synthesized at w4. Although there were traces of six anthocyanins at w4, the concentrations of the five types of cyanidin in RIL 1901 at b4 were 16–282 times higher than that at w4, while the concentration of peonidin 3-O-glucoside at w4 was similar to that at b4 (Table 1, Figure 1C). Thus, the remaining 14 anthocyanins and their derivatives contribute to the black root skin appearance at b4.

In addition to anthocyanins, 30 types of flavonoid metabolites were found only in RIL 1901 but not in RIL 1911. These flavonoid metabolites included one chalcone, two dihydroflavones, two dihydroflavonols, one dihydroisoflavone, 12 flavonoids, nine flavonols, one flavonoid carbonoside, one flavanol, and one proanthocyanidin. Kaempferol-3-O-(2″-galloyl)-β-D-galactopyranoside and kaempferol-3-O-(6″-galloyl)-β-D-glucopyranoside only accumulated at b4. Notably, chalcone and naringenin were detected at both b4 and w4 (Appendix A). The above results suggested that the upstream flavonoid biosynthesis pathway was the same at b4 and w4 but was blocked in the downstream of naringin at w4.

### 2.2. Transcriptome Analysis of Root Skin of Two Radish RILs

We further compared the changes in gene expression profiles among five root skin samples. Transcriptomic sequencing of 15 root skin samples yielded 117.78 Gb of clean data, with a Q30 base percentage of 92% or higher. A total of 73.91–82.76% of the clean reads were mapped to the radish reference genome (Appendix A). In the comparison of w4_Vs_b1, w4_Vs_b2, w4_Vs_b3, and w4_Vs_b4, 11,449 (up-regulated 7315, down-regulated 4134), 10,587 (up-regulated 5403, down-regulated 5184), 11,385 (up-regulated 5915, down-regulated 5470), and 5208 (up-regulated 3147, down-regulated 2061) significant differentially expressed genes (DEGs) were identified, respectively (Figure 2A, Appendix A). During the growth of radish, the number of up-regulated genes was larger than that of down-regulated genes. In addition, 1058 DEGs were up-regulated in the four comparisons, while 2995, 1007, 1811, and 446 DEGs were only up-regulated in w4_Vs_b1, w4_Vs_b2, w4_Vs_b3, and w4_Vs_b4 comparisons, respectively (Figure 2B). Meanwhile, 719 DEGs were down-regulated in the four comparisons, while 1181, 917, 1363, and 339 DEGs were down-regulated only in w4_Vs_b1, w4_Vs_b2, w4_Vs_b3, and w4_Vs_b4, respectively (Figure 2C).

### 2.3. Co-Expression Network Analysis Identified Flavonoid-Related DEGs

WGCNA has been frequently applied to analyze the data of transcriptome, proteome, and metabolome to identify feature-related co-expression modules [22]. In order to clarify the gene regulatory network of anthocyanin biosynthesis in the root skin of black radish, WGCNA was performed using filtered 22,963 DEGs (Appendix A). These DEGs were grouped into 13 modules, each of which was a highly related gene cluster (marked with different colors) (Figure 3A and Appendix A). The characteristic data of 16 anthocyanins and different developmental stages were used for module characteristic relationship analysis (Figure 3B,C). The correlation analysis of modules with anthocyanins and different developmental stages showed that the MEindianred2 module was highly positively correlated with most anthocyanins and with b3 (Figure 3B). It could be concluded that the genes in the MEindianred2 module played an important role in anthocyanin biosynthesis at b3, so the MEindianred2 module was selected for further research. Furthermore, GO enrichment analysis revealed that there were 48 secondary classifications in cell components, molecular functions, and biological processes. In the category of cell components, the genes in the MEindianred2 module were mainly related to cells, cell parts, and organelles. In terms of molecular function, most genes were enriched in binding, catalytic activity, and nucleic acid binding TF activity, as well as such biological processes as cellular processes, metabolic processes, and single biological processes (Figure 4A). To further determine the function of the genes in the MEindianred2 module, KEGG analysis was performed. KEGG enrichment analysis revealed that the genes in the MEindianred2 module were primarily linked to the secondary metabolite synthesis, plant-pathogen interaction, and flavonoid biosynthesis (Figure 4B). In addition, 16 structural genes (one *Rs4CL*, four *RsCHS*, three *RsCHI*, one *RsF3H*, one *RsF3*′*H*, one *RsDFR*, two *RsANS*, two *RsUGT75C1,* and one *RsUGT79B1*) related to anthocyanin biosynthesis were enriched in the Meindianred2 module, which further proved that the co-expressed genes in this module were involved in the accumulation of anthocyanins in the root skin of black radish.

### 2.4. TFs Involved in Flavonoid Synthesis

In the MEindianred2 module, 98 TFs were identified, including *AP2/ERF* (18), *WRKY* (13), *bHLH* (6), *MYB* (6), *NAC* (6), and *bZIP* (3) (Appendix A). Current studies have shown that a variety of TFs, especially *WRKY*, *MYB*, and *bHLH*, regulate anthocyanin biosynthesis. Therefore, we further analyzed *WRKY*, *MYB*, and *bHLH* and found that *WRKY44L* (2), *MYB114* (1), *MYB308L* (1), and *TT8* (1) were highly expressed at b3. It has been previously revealed that overexpression of *FaWRKY44* can significantly increase the anthocyanin concentration but decrease the proanthocyanin concentration in strawberry fruits. After *FaWRKY44* silencing by VIGS, the concentration of anthocyanidins is significantly decreased, but that of procyanidins is significantly increased in strawberry fruits [23]. *SmWRKY44* interacts with *SmMYB1* to promote anthocyanin biosynthesis in eggplant leaves [24]. These studies indicate that *WRKY* regulates anthocyanin biosynthesis through interaction with *MYB*. In addition, *MYB114* and *MYB308L* are regulatory factors for flavonoid synthesis and have been reported to be involved in anthocyanin biosynthesis. *MYB114* is up-regulated in four types of apples, suggesting that it may play an important role in regulating anthocyanin biosynthesis in red apples [25]. Moreover, *MdMYB308L* interacts with *MdbHLH33* and positively regulates anthocyanin accumulation in apples [26]. *TT8* plays a key role in anthocyanin biosynthesis in *Brassica*, and the MBW complex composed of *TT8* activates structural genes for late anthocyanin biosynthesis [27]. These findings indicate that most of the TFs are related to the biosynthesis of flavonoids and anthocyanins, further proving the close correlation between the genes in the MEindianred2 module and the synthesis of flavonoids and anthocyanins.

### 2.5. Candidate Hub Genes Related to Flavonoid Synthesis

To further determine the relationship between genes within the module and to filter the hub genes (highly connected genes), a correlation network was constructed. In the MEindianred2 module, the co-expression network was visualized in Cytoscape-free software using genes with WGCNA edge weight ≥ 0.38 (Appendix A). There were 193 highly correlated genes in total, including four phenylpropane biosynthesis genes, 16 flavonoid synthesis genes, and 17 TFs (Appendix A and Appendix A). UDP glycosyltransferase (*UGT75C1*), colorless anthocyanin dioxygenase (*LDOX*), glutathione S-transferase (*GST*), chalcone isomerase (*CHI*), UDP glycosyltransferase (*UGT84A2*), TMV resistance protein N, and chalcone synthase (*CHS*) exhibited the highest connectivity in the network (Appendix A). Many studies have demonstrated that anthocyanin biosynthesis is regulated by a variety of TFs. In order to better clarify anthocyanin biosynthesis, the interaction network diagram of highly connected structural genes and TFs was reconstructed. In the network diagram, a total of ten TFs were highly correlated with 16 structural genes (Figure 5). Among the ten TFs, LOC108840410 (*RsMYB114*), LOC108832389 (*RsMYB308L*), LOC108838184 (*RsTT8*), and LOC108806342 (*RsWRKY44L*) had the highest connectivity with 16 structural genes. Among the 16 structural genes, LOC108814337 (*RsCHS*), LOC108814129 (*RsCHI*), LOC108826061 (*RsDFR*) and LOC108821079 (*RsUGT75C1*) had the highest connectivity with ten TFs (Figure 5). The expression levels of the above four TFs and four structural genes began to be up-regulated at b1, peaked at b3, then were down-regulated at b4, and almost turned out to be zero at w4. Previous studies have shown that *MYB114*, *MYB308L*, *TT8*, and *WRKY44L* are positive regulators of anthocyanin biosynthesis. Therefore, it is inferred that they may promote the accumulation of anthocyanins, thus making radish root skin black. These findings further indicate that the MEindianred2 module is related to anthocyanin biosynthesis.

### 2.6. Results of the Transcriptome and Metabolome Comprehensive Analyses

To better understand the relationship between genes and metabolites and to predict the coloring mechanism in black radish, the anthocyanin biosynthesis pathway in the root skin of black radish was constructed by integrative analysis of the expression of genes related to flavonoid and anthocyanin biosynthesis pathways (Ko00941 and Ko00942) and the relative expression of metabolites (Figure 6). Three groups of genes showed unique expression patterns during coloring in black radish. In the first group, genes included early structural genes of the flavonoid biosynthesis pathway, such as *RsPAL* and *RsC4H*, which remained active at the entire growth stage of radish, as well as at w4. In the second group, *RsFLS* was the only gene, and its expression at b1 was the highest. In the third group, genes included *Rs4CL*, *RsCHS*, *RsCHI*, *RsF3H* (except LOC108805892), *RsF3*′*H*, *RsDFR*, and *RsANS*, whose expressions began to be up-regulated at b1, peaked at b3, then were down-regulated at b4 and became very low or were almost absent at w4. The concentrations of naringin and naringin chalcone were the lowest at w4, which also implied that the difference in root skin coloring may start from flavonoid biosynthesis. Furthermore, *DFR* is bound to dihydroquercetin but less to other substrates, which may lead to the gradual decrease of dihydroquercetin concentration. *RsFLS* was more active at b1, resulting in the sudden increase of kaempferol, quercetin, and myricetin at b2. Furthermore, the high expressions of *RsUGT75C1* and *RsUGT79B1* at b3 made the relative concentrations of cyanidin derivatives and paeoniflorin derivatives the highest at b4, suggesting that the metabolites are synthesized and accumulate due to gene expression. In addition, among the 28 structural genes of the flavonoid and anthocyanin biosynthesis pathway (Ko00941 and Ko00942), 20 structural genes were included in the MEindianred2 module, which further confirmed that the MEindianred2 module was closely related to anthocyanin biosynthesis.

### 2.7. Verification of RNA-Seq Results by qRT-PCR

A total of 12 DEGs, including nine structural genes and three TFs, were selected to verify the reliability of the results of RNA-Seq. The expression levels of these genes at b1, b2, b3, b4, and w4 were analyzed by qRT-PCR (Figure 7). The results showed that the expression trend of these genes was consistent with that of the sequencing results, indicating that the RNA data are valid and reliable.

## 3. Discussion

At present, food rich in anthocyanins is the focus of consumers and researchers. In this study, UPLC-MS/MS and RNA-seq were used to analyze the metabolites and transcriptome of the root skin of radish RIL 1901 at four stages and line RIL 1911 at the maturity stage. The analysis of differences in metabolite accumulation in the ripening process of black radish revealed that flavonoids were the main metabolites of root skin coloring, and the flavonoids accumulating in the root skin of black radish at different stages in the ripening process were determined, including 12 metabolites such as flavonols, flavonoids, and anthocyanins. In cruciferous vegetable crops, the basic component of purple pakchoi anthocyanin was cyanidin-malonyl-glucoside [28]. 17 types of anthocyanins were detected and identified in purple Tsai-tai and the leaves of purple pakchoi, all of which were cyanidin. The anthocyanins in purple cabbage were mainly mustard acylation of cyanidin-3-sophoricide-5-glucoside, while there were coumarin or ferulic derivatives of cyanidin-3-sophoricide-5-glucoside in purple cauliflower [29]. Six anthocyanins were detected in the petiole of purple turnip, the aglycones were pelargonidin, and the petiole was brick red [30]. However, previous studies have confirmed that the anthocyanins in the red root skin or flesh of red radish, carmine radish, and ‘xinlimei’ radish are mainly pelargonidin [13,31,32], which indicates that the anthocyanins responsible for red coloring in radish are mainly pelargonidin. In our study, a total of 16 anthocyanins were detected in the root skin of radish, including 12 cyanidins, one pelargonidin, and three peonidins. The relative concentration of cyanidin 3-o-glucoside was the highest. Therefore, the accumulation of cyanidin and its derivatives resulted in the black root skin of radish. The color of delphinidin ranges from magenta to purple [33], and delphinidin and its derivatives were not detected in the root skin of black radish.

Only differential metabolites can be detected by metabolomic analysis, and the changes in the genes and the proteins are unclear. At present, many studies have combined the analysis of transcriptome, proteome, and metabolome. In order to study the relationship between the composition and concentration of anthocyanins and the expression of structural genes and TFs related to the anthocyanin biosynthesis pathway, we concatenated the DEGs with 16 anthocyanins and five stages by WGCNA and identified a module positively related to anthocyanin accumulation. Within the module, structural genes (*RsCHS*, *RsCHI*, *RsDFR*, and *RsUGT75C1*) and TFs (*RsTT8*, *RsWRKY44L*, *RsMYB114*, and *RsMYB308L*) may play an important role in the anthocyanin biosynthesis in the root skin of black radish. The anthocyanin biosynthesis pathway in the root skin of black radish was constructed by integrative analysis of the expression of genes related to flavonoid and anthocyanin biosynthesis pathways (Ko00941 and Ko00942) and the relative expression of metabolites. The synthesis of delphinidin depends on the expression of *F3*′*5*′*H*. We did not detect *F3*′*5*′*H* in our transcriptome data, which indicated that the anthocyanin biosynthesis in the black root skin of radish involves no delphinidin-based anthocyanins. Located in the middle and downstream of the anthocyanin biosynthesis pathway, *DFR* catalyzes dihydroflavonol to form unstable proanthocyanidins [34]. In this study, it was found that *DFR* may specifically bind to dihydroquercetin but less to other substrates, resulting in a gradual reduction of dihydroquercetin concentration. An amino acid substitution analysis has demonstrated that *DFR* can selectively catalyze amino-related substrates [35]. *UGT75C1*, the last enzyme in anthocyanin biosynthesis, converts unstable anthocyanins into stable anthocyanins through glycosylation, which is the key reaction in the last step of anthocyanin biosynthesis [36]. Interestingly, almost all of our key structural genes were highly expressed at b3, while the relative concentrations of cyanidin derivatives and paeoniflorin derivatives were the highest at b4, implying that the metabolites are synthesized and accumulated due to gene expression.

The plant anthocyanin biosynthesis is not only controlled by structural genes but also affected by regulatory genes and other factors. At present, *MYB*, *bHLH,* and *WD40* are the TFs related to the anthocyanin biosynthesis of plants that have attracted the most attention. They can not only form a ternary complex MBW to regulate the accumulation of anthocyanins in plants but also independently regulate the accumulation of anthocyanins in plants [37,38,39]. In this study, annotation genes corresponding to *WD40* were not found in all DEGs. Previous studies have illustrated that the overexpression of *MdMYB114* leads to the accumulation of anthocyanins in apple callus. *MdMYB114* does not form a MBW complex, but it promotes the biosynthesis and transportation of anthocyanins by directly binding to the promoters of *MdANS*, *MdUFGT*, and *MdGST* so as to facilitate the expressions of these genes [40]. *MdMYB308L* interacts with *MdbHLH33* and plays a positive regulatory role in the anthocyanin accumulation of apples [26]. The co-expression of *RsMYB1* and *RsTT8* induces anthocyanin accumulation and up-regulates the expression of genes in the anthocyanin biosynthesis pathway of radish [17]. The above studies confirm that MYB regulates anthocyanin biosynthesis alone or by interacting with *bHLH*. Therefore, we speculated that *MYB* interacts with *bHLH* to regulate anthocyanin biosynthesis in the root skin of black radish. In addition, *WRKY* regulates anthocyanin biosynthesis by interacting with *MYB* or *bHLH* complexes, and *SmWRKY44* interacts with *SmMYB1* to promote the anthocyanin biosynthesis of eggplant leaves [24]. Hence, we speculated that *RsMYB114*, *RsMYB308L*, *RsTT8*, and *RsWRKY44L* play regulatory roles in the anthocyanin biosynthesis in the root skin of black radish.

## 4. Materials and Methods

### 4.1. Plant Materials

A black RIL (RIL 1901) and a white RIL (RIL 1911) with white root flesh produced by the radish research group of the College of Horticulture, Shanxi Agricultural University, were selected as the research materials. The root skin color of RIL 1901 gradually changed to black, while that of RIL 1911 was always white (Figure 1A).

For RIL 1901, the roots of 10 plants after 15 days of sowing were mixed into one biological replicate and labeled as b1 (seedling stage); the root skins of 6 plants after 30 days of sowing were mixed into one biological replicate and labeled as b2 (cortical division stage); the root skins of 3 plants after 45 days of sowing were mixed into one biological replicate and labeled as b3 (rapid expansion stage); the root skins of 3 plants after 60 days of sowing were mixed into one biological replicate and labeled as b4 (maturity stage). For RIL 1911, the root skins of 3 plants after 60 days of sowing were mixed into one biological replicate and labeled as w4 (maturity stage) (Figure 1A). Three biological replicates were set for each sample. Immediately after sampling, these roots were frozen in liquid nitrogen and stored at −80 °C for subsequent metabolite extraction, transcriptome sequencing, and qRT-PCR analysis.

### 4.2. Flavonoid Identification and Quantification

The sample preparation, extraction analysis, metabolite identification, and quantitative analysis were conducted at Metware Biotechnology Co., Ltd. (Wuhan, China) following their standard procedures and previously fully described by Yuan et al. [41]. Specifically, flavonoid extracts were analyzed using ultra-high performance liquid chromatography-tandem mass spectrometry (UPLC-MS/MS) (Shim-pack UFLC SHIMADZU CBM30A, Applied Biosystems 4500 QTRAP, Waltham, MA, USA). Then, metabolite quantification was performed by multiple reaction monitoring (MRM) using a triple quadrupole mass spectrometer. Later, the identified metabolites were subjected to orthogonal partial least-squares discriminate analysis (OPLS-DA), and metabolites with |log2FC| ≥ 1, *p* < 0.05, and variable importance in projection (VIP) score ≥ 1 were considered as differentially accumulated flavonoids (DAFs).

### 4.3. RNA-Seq and Differential Expression Analysis

The extraction and sequencing of the total RNA were performed by Metware Biotechnology Co., Ltd. (Wuhan, China), and clean reads were mapped to the *Raphanus sativus* using HISAT2 [42]. FPKM was used for gene/transcript level quantification. Based on the raw count data, differential expression analysis between samples was carried out by the DESeq2 R package (1.16.1) [43]. Genes with |log2FC| ≥ 1, and a false discovery rate of (FDR) < 0.05 were defined as DEGs and subjected to Gene Ontology (GO) enrichment analysis and Kyoto Encyclopedia of Genes and Genomes (KEGG) analysis.

### 4.4. Co-Expression Network Analysis for the Construction of Modules

For co-expression network analysis, the WGCNA software package was used [22]. In order to obtain genes related to anthocyanin biosynthesis, DEGs detected at b1, b2, b3, b4, and w4 were selected and comprehensively analyzed with 16 anthocyanins and 5 stages, respectively. Subsequently, KEGG enrichment analysis and GO enrichment analysis were performed. The co-expression network was visualized by Cytoscape.

### 4.5. Comprehensive Analysis of the Transcriptome and Metabolome

To better understand the relationship between genes and metabolites and to predict the molecular coloring mechanism in black radish, the key DEGs and DAFs were filtered by biological function analyses such as flavonoid and anthocyanin biosynthesis pathways (Ko00941 and Ko00942) and the correlation analysis. Later, the obtained DEGs and DAFs were simultaneously mapped to KEGG pathways to construct the anthocyanin biosynthesis pathway in the root skin of black radish.

### 4.6. QRT-PCR

To validate the RNA-seq results, 10 structural genes and 4 transcription factors (TFs) involved in the anthocyanin biosynthesis were subjected to qRT-PCR using gene-specific primers (Appendix A). These gene-specific primers were designed using Primer 3 software according to the gene sequence. The expression level of the target gene was determined with the EF-1-α gene (GenBank login number: GO479260) as an internal reference. The total RNA of each sample was extracted at b1, b2, b3, b4, and w4 according to the TRIzol Reagent User Guide (Life Technologies, Carlsbad, CA, USA). The first-strand cDNA was obtained using the PrimeScript^TM^ RT Master Mixreverse transcription Kit (Takara, Dalian, China). QRT-PCR was repeated three times in a 20 μL reaction system containing 10 μL of 2 × TB Green Premix EX Taq, 2.0 μL of diluted cDNAs, 0.4 μL of 50 × ROX Reference Dye II, 0.4 μL of forward primers (10 μM), 0.4 μL of reverse primers (10 μM), and sterile ddH_2_O. PCR was carried out in an ABI 7500 instrument (ABI, Oakland, CA, USA), and the thermal cycling parameters were as follows: 30 s at 94 °C, followed by 40 cycles (30 s at 95 °C, 30 s at 60 °C, and 30 s at 72 °C). The relative gene expression level was calculated using the 2^−ΔΔCT^ method [44].

## 5. Conclusions

In this study, the metabolomics and transcriptomics analyses of flavonoids in RILs 1901 and 1911 were performed to clarify the coloring mechanism in black radish. The expression levels of genes related to anthocyanin biosynthesis were significantly higher in RIL 1901 than in RIL 1911. Meanwhile, a module positively related to anthocyanin accumulation was identified by WGCNA. Within the module, structural genes (*RsCHS*, *RsCHI*, *RsDFR*, and *RsUGT75C1*) and TFs (*RsTT8*, *RsWRKY44L*, *RsMYB114*, and *RsMYB308L*) may play an important role in anthocyanin biosynthesis in the black root skin of radish. This study not only provides new insights into the anthocyanin biosynthesis and accumulation in the root skin of black radish but also lays a molecular foundation for accelerating the cultivation of new black radish varieties.

## Figures and Tables

**Figure 1 ijms-24-13663-f001:**
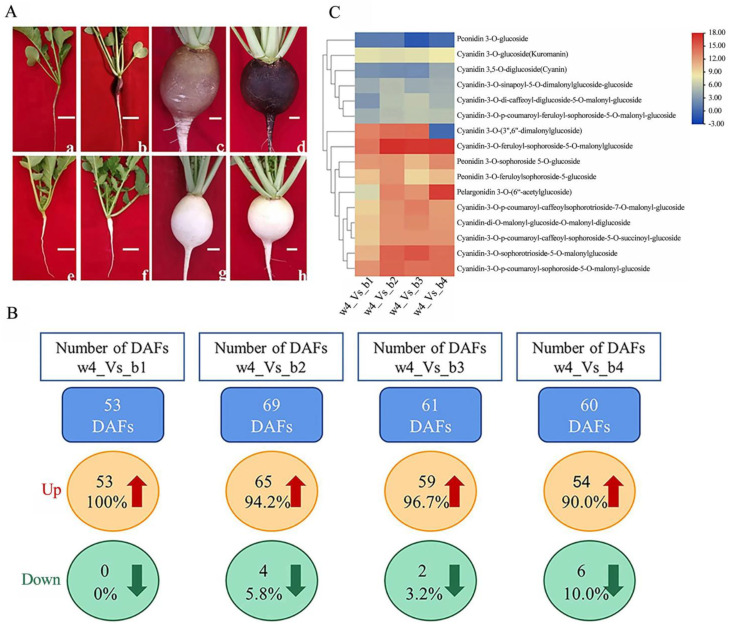
Fleshy root phenotype of two RILs at different growth stages and number of DAFs between two RILs. (**A**) Fleshy root phenotype of two RILs at different growth stages; (**B**) number of DAFs between two RILs; (**C**) heat map of sixteen anthocyanins. a, b, c, and d are the phenotypes of inbred line 1901 after 15, 30, 45, and 60 days of direct seeding; e, f, g, and h are the phenotypes of inbred line 1911 after 15, 30, 45 and 60 days of direct seeding. The root skin of inbred line 1901 after 15, 30, 45, and 60 days of direct seeding was renamed b1, b2, b3, and b4, respectively. The root skin of inbred line 1911, after 60 days of direct seeding, was renamed as w4.

**Figure 2 ijms-24-13663-f002:**
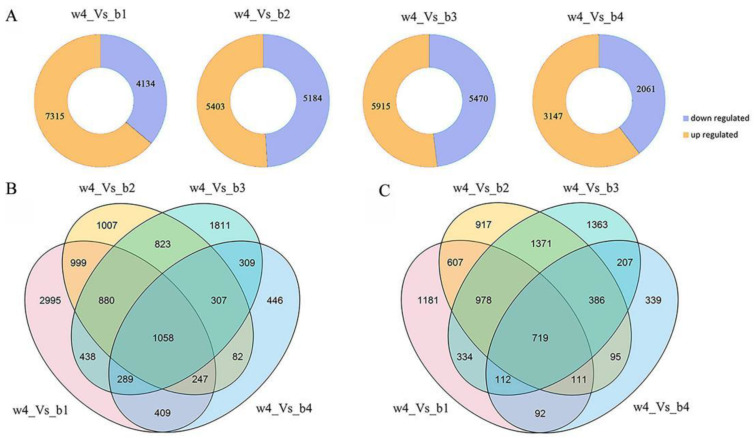
Statistics of the number of DEGs. (**A**) Statistics on the quantity of DEGs up regulation and down regulation in the comparison of the four groups; (**B**) Venn diagram describes the total number and unique number of up-regulated DEGs in the four groups of comparison; (**C**) Venn diagram describes the total number and unique number of down-regulated DEGs in the four groups of comparison.

**Figure 3 ijms-24-13663-f003:**
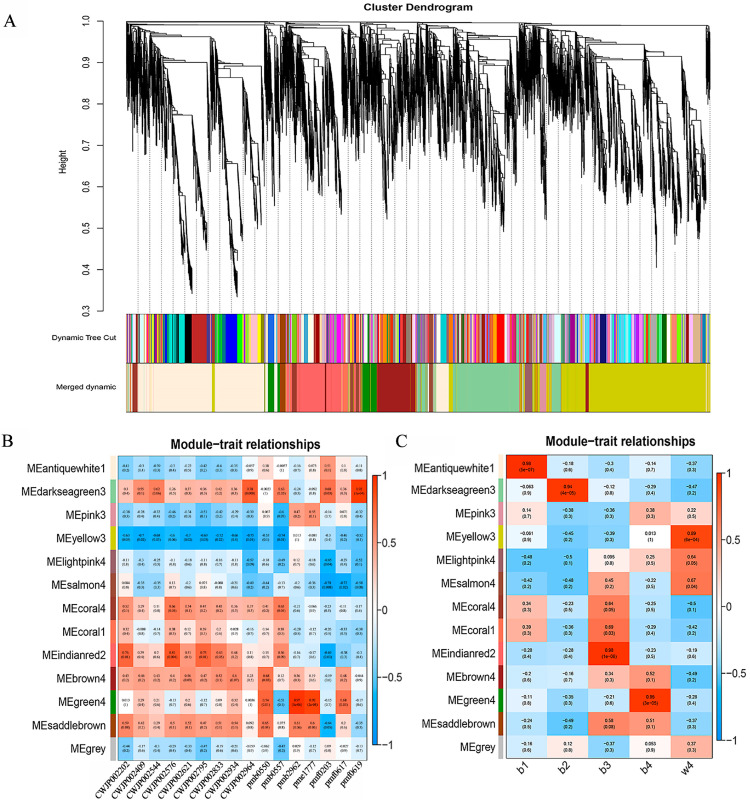
Weighted gene co-expression network analysis (WGCNA) of 22,963 DEGs. (**A**) Hierarchical clustering tree showing 13 modules of co-expressed genes by WGCNA. Each leaf of the tree corresponds to one gene. The major tree branches constitute 13 modules, labeled with different colors; (**B**) module—anthocyanin relationship. Each row represents a module, and each column represents a specific anthocyanin (detailed information reference Table 1). The value in each cell at the row–column intersection represents the correlation coefficient between the module and the anthocyanin and is displayed according to the color scale on the right. The value in parentheses in each cell represents the *p* value; (**C**) modules—five periods association.

**Figure 4 ijms-24-13663-f004:**
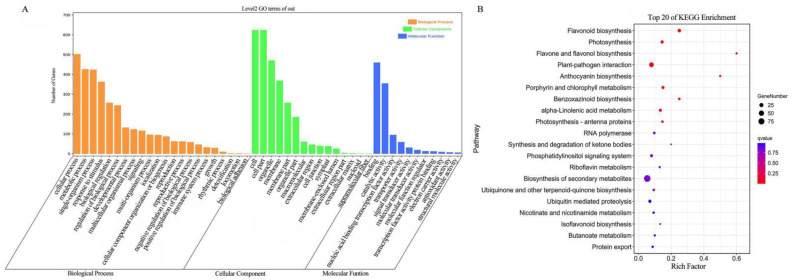
GO and KEGG analysis of genes in the MEindianred2 module. (**A**) GO enrichment analysis of DEGs in the MEindianred2 module; (**B**) KEGG pathway enrichment analysis of DEGs in the MEindianred2 module.

**Figure 5 ijms-24-13663-f005:**
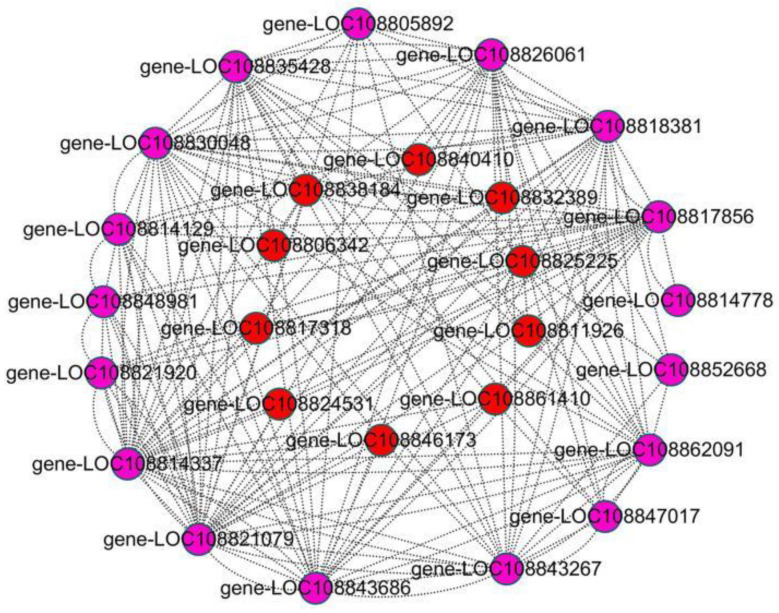
Co-expression network of sixteen structural genes and ten transcription factors in the MEindianred2 module. Red—transcription factors; purple—structural genes.

**Figure 6 ijms-24-13663-f006:**
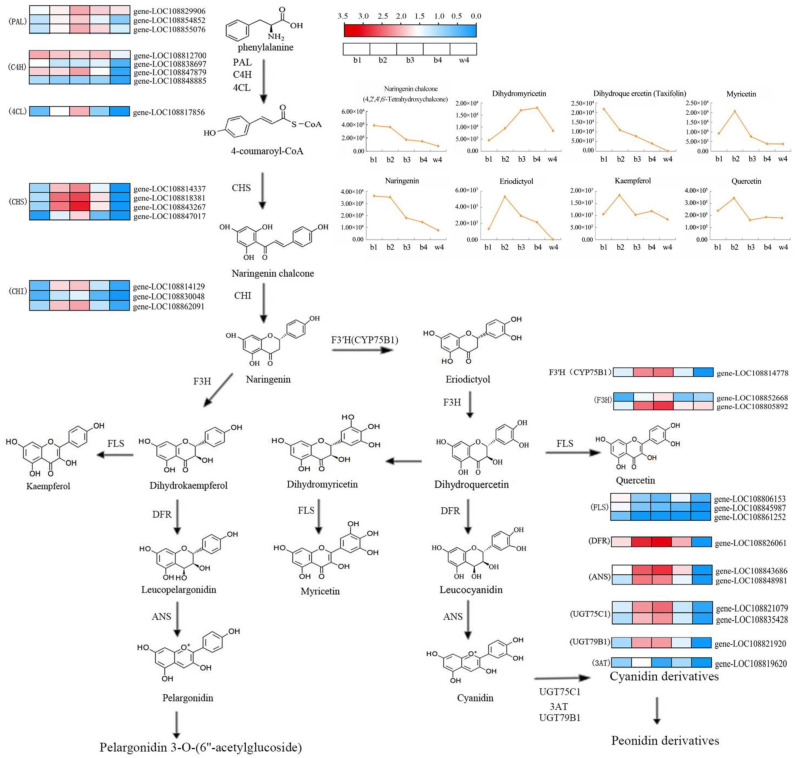
The pathway of anthocyanin biosynthesis in black skin of radish. Heat maps are shown for changes in the expression level of genes related to anthocyanidin synthesis. Line charts are shown for changes in the metabolite content in anthocyanidin synthesis pathways. PAL, phenylalanine aminotransferase; C4H, cinnamate hydroxylase; 4CL, p-Coumarate CoA Ligase; CHS, chalcone synthase; CHI, chalcone isomerase; F3H, flavonoid 3-hydroxylase; F3′H, flavonoid 3′-hydroxylase; FLS, flavonol synthase; DFR, dihydroflavonol 4-reductase; ANS, anthocyanidin synthase; UGT, UDP glycosyl-transferase.

**Figure 7 ijms-24-13663-f007:**
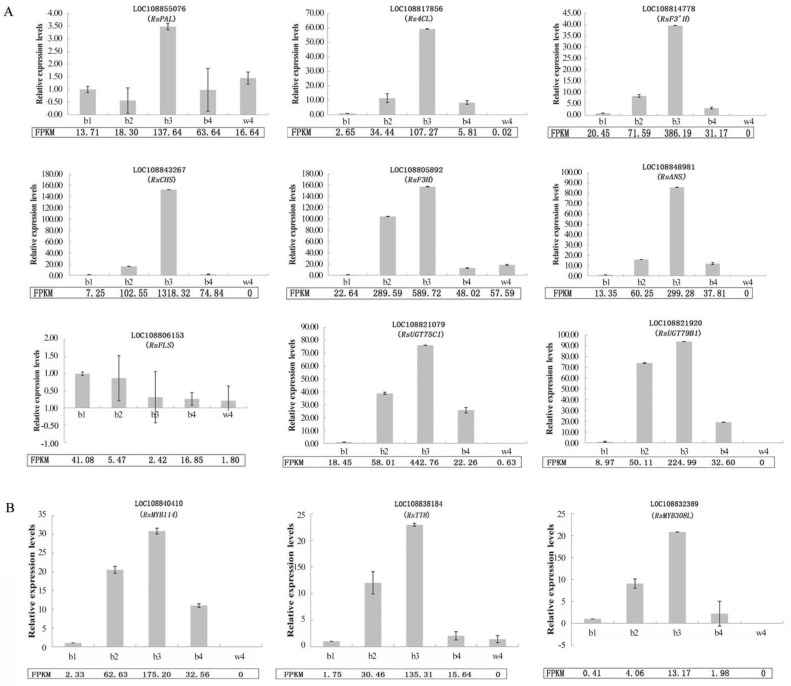
qRT-PCR validation of RNA-seq results based on gene expression level. (**A**) Nine structural genes; (**B**) three TFs. FPKM: fragments per kilobase of transcript per million fragments mapped. Each bar represents the average of three biological replicates plus the standard deviation.

**Table 1 ijms-24-13663-t001:** Differentially accumulated anthocyanins during the ripening process in radish.

Meta ID	Metabolite Name	Ion Abundance				
		b1	b2	b3	b4	w4
CWJP002202	Cyanidin-3-O-sophorotrioside-5-O-malonylglucoside	1.83 × 10^4^	2.70 × 10^5^	4.45 × 10^5^	1.99 × 10^5^	9.00
CWJP002409	Cyanidin-3-O-p-coumaroyl-sophoroside-5-O-malonyl-glucoside	5.43 × 10^4^	2.58 × 10^5^	1.75 × 10^5^	1.88 × 10^5^	9.00
CWJP002544	Cyanidin-3-O-feruloyl-sophoroside-5-O-malonylglucoside	1.39 × 10^5^	1.67 × 10^6^	1.31 × 10^6^	1.35 × 10^6^	9.00
CWJP002576	Cyanidin-3-O-di-caffeoyl-diglucoside-5-O-malonyl-glucoside	4.74 × 10^3^	2.93 × 10^4^	4.06 × 10^4^	1.77 × 10^4^	6.70 × 10^2^
CWJP002621	Cyanidin-3-O-p-coumaroyl-feruloyl-sophoroside-5-O-malonyl-glucoside	1.62 × 10^4^	3.25 × 10^4^	4.13 × 10^4^	3.57 × 10^4^	7.88 × 10^2^
CWJP002795	Cyanidin-3-O-p-coumaroyl-caffeoylsophorotrioside-7-O-malonyl-glucoside	6.55 × 10^3^	5.84 × 10^4^	1.11 × 10^5^	3.96 × 10^4^	9.00
CWJP002833	Cyanidin-di-O-malonyl-glucoside-O-malonyl-diglucoside	9.51 × 10^3^	5.50 × 10^4^	7.36 × 10^4^	4.41 × 10^4^	9.00
CWJP002934	Cyanidin-3-O-p-coumaroyl-caffeoyl-sophoroside-5-O-succinoyl-glucoside	8.28 × 10^3^	5.17 × 10^4^	5.02 × 10^4^	5.17 × 10^4^	9.00
CWJP002964	Cyanidin-3-O-sinapoyl-5-O-dimalonylglucoside-glucoside	6.52 × 10^3^	1.49 × 10^4^	6.72 × 10^3^	8.07 × 10^3^	3.62 × 10^2^
pmb0550	Cyanidin 3-O-glucoside (Kuromanin)	2.51 × 10^6^	1.90 × 10^6^	2.25 × 10^6^	3.69 × 10^6^	1.31 × 10^4^
pmb0557	Cyanidin 3-O-(3″,6″-dimalonylglucoside)	9.20 × 10^4^	1.97 × 10^5^	2.17 × 10^5^	9.00	9.00
pmb2962	Pelargonidin 3-O-(6″-acetylglucoside)	1.12 × 10^3^	9.26 × 10^4^	4.99 × 10^4^	1.02 × 10^6^	9.00
pme1777	Cyanidin 3,5-O-diglucoside (Cyanin)	2.22 × 10^3^	1.67 × 10^3^	1.56 × 10^3^	5.70 × 10^3^	3.43 × 10^2^
pmf0203	Peonidin 3-O-glucoside	1.03 × 10^6^	9.70 × 10^5^	1.99 × 10^5^	4.91 × 10^5^	4.37 × 10^5^
pmf0617	Peonidin 3-O-sophoroside 5-O-glucoside	4.15 × 10^4^	5.44 × 10^4^	1.43 × 10^4^	6.76 × 10^4^	9.00
pmf0619	Peonidin 3-O-feruloylsophoroside-5-glucoside.	1.07 × 10^4^	5.58 × 10^4^	5.96 × 10^3^	1.36 × 10^4^	9.00

## Data Availability

Not applicable.

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
