# Peer review of "Molecular Regulatory Network of Anthocyanin Accumulation in Black Radish Skin as Revealed by Transcriptome and Metabonome Analysis"

_ijms, 2023, doi:10.3390/ijms241713663_

Round 1
Reviewer 1 Report
The authors: Jing Zhang et al. report a study on “Molecular regulatory network of anthocyanin accumulation in black radish skin as revealed by transcriptome and metabonome analysis”.
In this study, metabolomic and transcriptomic analysis of flavonoids in the root skin of two radishes at different development stages, is matched to clarify the coloring mechanism in black radish
The study is well defined and the structure is highly articulated
Just a few notes:
To make the identification and quantification of flavonoids by LC-MS more comprehensive, the authors should expand the discussion of materials and methods, or include a reference.
Page 13, line 417: “Tab 10” should be correct to “Tab S10”
Author Response
We have revised the manuscript after careful consideration of the comments made by the two reviewers and editor. Here are our responses in detail. The revised parts have been highlighted in yellow.
Reviewer #1
In this study, metabolomic and transcriptomic analysis of flavonoids in the root skin of two radishes at different development stages, is matched to clarify the coloring mechanism in black radish.
The study is well defined and the structure is highly articulated.
Response: We are very grateful to the reviewer for your affirmation and hard work.
- To make the identification and quantification of flavonoids by LC-MS more comprehensive, the authors should expand the discussion of materials and methods, or include a reference.
Response: Your proposal is reasonable. We have added a reference for clarity flavonoid identification and quantification. And, page 12, 4.2 has been corrected and highlighted in yellow. Meanwhile, we have added this reference to the list of references.
- Page 13, line 417: “Tab 10” should be correct to “Tab S10”
Response: We carefully checked and revised the whole manuscript. The revised parts have been highlighted in yellow.

Reviewer 2 Report
The work has simple design but has a clear aim and brings new data. I have few comments for improvement:
- I am not an expert on the transcriptome, so an expert must comment on this.
- in terms of metabolites, I appreciate the large variety of substances identified, however, from the methodological point of view, it is not clear how the identification was done, please add (or also a technical citation of your other work, if you have already applied this method).
- Line 89 and following: the sentence "A total of 172 types of flavonoids were detected, including 63 flavonols, 51 flavonoids (39 flavonoids and 12 flavonoid carbonosides)" is not clear, 51 and 39 flavonoids mean what? flavonoid is superimposed for the whole group, correct/modify
- if I see correctly, the quantification was implemented only as a relative content, is it a problem to determine the absolute content (w/g DW) of at least major compounds that have available commercial standards? This would be desirable for comparison with other plants and would certainly increase the quality of the results (see e.g. https://www.sciencedirect.com/science/article/abs/pii/S0981942822002819?via%3Dihub)
- I also recommend quantifying at least the total content of phenols and possibly anthocyanins spectrophotometrically, which will allow comparison with other crops and increase the citation potential of the manuscript
- numbers/letters in fig. 6 and 7 are hard to read, use a larger font and possibly enlarge the image over the entire width of the page
Author Response
We have revised the manuscript after careful consideration of the comments made by the two reviewers and editor. Here are our responses in detail. The revised parts have been highlighted in yellow.
Reviewer #2
- The work has simple design but has a clear aim and brings new data.
Response: Thank you very much for positive judgment of our work.
- I am not an expert on the transcriptome, so an expert must comment on this.
Response: There is another reviewer, so you don't have to worry about.
- in terms of metabolites, I appreciate the large variety of substances identified, however, from the methodological point of view, it is not clear how the identification was done, please add (or also a technical citation of your other work, if you have already applied this method).
Response: Your proposal is reasonable. We have added a reference for clarity flavonoid identification and quantification. And, page 12, 4.2 has been corrected and highlighted in yellow. Meanwhile, we have added this reference to the list of references.
- Line 89 and following: the sentence "A total of 172 types of flavonoids were detected, including 63 flavonols, 51 flavonoids (39 flavonoids and 12 flavonoid carbonosides)" is not clear, 51 and 39 flavonoids mean what? flavonoid is superimposed for the whole group, correct/modify.
Response: We carefully checked and revised the manuscript. Page 2, 2.1 has been revised and highlighted in yellow.
- if I see correctly, the quantification was implemented only as a relative content, is it a problem to determine the absolute content (w/g DW) of at least major compounds that have available commercial standards? This would be desirable for comparison with other plants and would certainly increase the quality of the results (see e.g. https://www.sciencedirect.com/science/article/abs/pii/S0981942822002819?via%3Dihub)
Response: Many thanks for your suggestion. Yes, the metabolomics data is a relatively quantitative result aimed at detecting differences in metabolite contents between different varieties or different treatments. Because we were not conducting research on specific metabolite, we did not examine the absolute quantification. Thank you very much for your suggestion. In future research, we will perform the absolute quantitative analysis of metabolites of interest with available commercial standards.
- I also recommend quantifying at least the total content of phenols and possibly anthocyanins spectrophotometrically, which will allow comparison with other crops and increase the citation potential of the manuscript.
Response: Your suggestion is very good, but we do not have stored samples. The planting of materials and the measurement of data will take at least 62 days, so we cannot supplement the data in a short space of time.
- numbers/letters in fig. 6 and 7 are hard to read, use a larger font and possibly enlarge the image over the entire width of the page
Response: To make the manuscript easier to read, we carefully checked and revised the numbers/letters in the figures of manuscript. All the figures in the manuscript are revised images.
